# Cost analysis of the TB-PRACTECAL clinical trial on novel tuberculosis treatment regimens

Dzintars Gotham[1‡], Manuel Martin[1‡], Melissa J Barber[1,2], Emil Kazounis[3], Charlotte Batts[3], Rosalind Scourse[1*], Greg Elder[1], Bern-Thomas Nyang'wa[4]

1 MSF Access Campaign, Geneva, Switzerland, 2 Yale Collaboration for Regulatory Rigor, Integrity, and Transparency (CRRIT), New Haven, United States of America, 3 MSF UK, London, United Kingdom, 4 MSF Operational Centre Amsterdam, Amsterdam, Netherlands

‡ These authors are joint first authors on this work.
* roz.scourse@london.msf.org

## Abstract

Clinical trials are considered to be the largest contributor to pharmaceutical development costs. However, public disclosure of the costs of individual clinical trials is rare. Médecins Sans Frontières (MSF) sponsored a phase 2b-3 randomised controlled trial (TB-PRACTECAL), which identified a new treatment regimen for drug-resistant TB. We aimed to analyse the costs of undertaking a pivotal clinical trial conducted in relatively low-resource health settings and to demonstrate the feasibility of reporting clinical trial costs. TB-PRACTECAL trial costs were analysed using MSF accounting documents. Costs were broken down by cost category, year, and trial site. Total costs for TB-PRACTECAL were €33.9 million and the average cost per patient was €61,460. Twenty-six percent of total costs represented central activities (e.g. trial planning, trial management) and 72% represented trial site activities, with 2% uncategorizable. Within trial site costs, personnel costs were the largest cost (43%) followed by external diagnostic services (11%), medicines (9%), and other medical consumables (7%). Cost variation across trial sites was driven by different varying levels of pre-existing trial infrastructure. A review of previous studies yielded a wide range of cost estimates for clinical trials (ranging US$7–221 million/trial for pharmaceutical phase 2 and 3 trials). Nearly all previous estimates derive from industry reporting that is neither standardized nor auditable; to our knowledge, this is the first published comprehensive analysis of direct expenditures of a specific clinical trial including detailed cost breakdowns. The €34 million cost of TB-PRACTECAL included investments in developing clinical trial infrastructure, the complexity of managing six sites across three health systems, and medical expenditures that are not typical of standard clinical trials. Greater transparency in drug development costs can inform medicine pricing negotiations and is a key element in the design and implementation of more equitable systems of biomedical research and development.

**Data availability statement:** The data supporting the findings of this study are available in the Figshare repository (DOI: https://doi.org/10.6084/m9.figshare.25707459).

**Funding:** This study was funded by Médecins Sans Frontières (https://www.msf.org/) through a grant to DG, MM, and MJB. MJB, EK, CB, RS, GE, and B-TN are employees of Médecins Sans Frontières, and played roles in the study design, data collection and analysis, decision to publish, and preparation of the manuscript, as described in the author contributions. MSF had control over the final decision to publish.

**Competing interests:** DG, MM, and MJB received payments in the form of consultancy fees for this analysis. DG has received funding from Médecins Sans Frontiéres, the World Health Organization, Unitaid, the World Intellectual Property Organization, Medicines Patent Pool, Treatment Action Group, Global Justice Now, STOPAIDS, CoLab International, and the Ada Lovelace Institute, and declares no competing interests. MJB has received funding from Médecins Sans Frontières, the World Health Organization, the Ada Lovelace Institute, and Yale University, and declares no competing interests. MM has received funding from Médecins Sans Frontières, Drugs for Neglected Disease Initiative, STOPAIDS, and the Oxfam International, and declares no competing interests. MJB, EK, CB, RS, GE, and B-TN are employees of Médecins Sans Frontières and declare no competing interests. The specific roles of these authors are articulated in the 'author contributions' section. There are no patents, products in development or marketed products to declare. This does not alter our adherence to PLOS Global Public Health policies on sharing data and materials.

## Introduction

There were an estimated 410,000 new cases of drug-resistant TB (DR-TB) in 2022 [1]. DR-TB is significantly harder to treat than drug-sensitive TB, with treatment success rates of 50-75% versus over 85%, respectively [1]. Despite the significant disease burden and mortality, DR-TB has historically been a neglected area for pharmaceutical research investment. When bedaquiline was conditionally approved in 2012, it marked the end of a 50-year innovation drought during which no new TB medicines were developed and clinicians had to rely on repurposing older anti-microbials with burdensome side-effects for the treatment of DR-TB [2]. This was followed by the approvals of delamanid in 2014 and pretomanid in 2017. While TB requires treatment with a combined regimen of several medicines [3], there was limited to no evidence on how these new medicines could be integrated in treatment regimens. In the decade since their approval, several clinical trial programs have searched for the most effective, safe, shorter all oral treatment regime [4].

Médecins Sans Frontières (Doctors Without Borders; MSF), an international humanitarian organization, undertook a multicenter two-stage adaptive phase 2b-3 randomized controlled trial, TB-PRACTECAL, testing a new short, effective, and safe treatment regimen for drug-resistant TB. The trial was conducted in Belarus, South Africa, and Uzbekistan (see S1 Text for a list of trial sites). The trial enrolled 552 patients across 6 sites, with the protocol including 22 planned clinic visits over 108 weeks, alongside 3 sub-studies [5]. The trial was delivered with 19 partner organizations. Including planning, the project lasted 10 years.

In the trial, completed in 2021, a 6-month, all-oral regimen of bedaquiline, pretomanid, linezolid, and moxifloxacin (BPaLM) was statistically superior to the WHO standard of care of 9-to-24 months of treatment, with a favorable outcome in 88% of patients with BPaLM versus 59% with the standard of care [5]. Patients reported faster improvement in symptoms, and greater satisfaction the study regimens in comparison to the standard of care [6]. The results of the TB-PRACTECAL trial led to WHO endorsement of the BPaLM regimen as the recommended first-line treatment for rifampicin-resistant or multidrug-resistant TB [4]. The trial included three sub-studies on cost-effectiveness, quality of life endpoints, and pharmacokinetics and pharmacodynamics (pending publication) [7–9].

Clinical trial costs are largely undisclosed [10], and the data that are available provides only high-level average costs or cost estimates, rather than detailed costs for individual trials [11–13]. There have been growing calls for increased transparency in the costs of research and development (R&D) [14–16], as increased understanding of these costs could inform the design of medicines pricing policies, innovative R&D financing mechanisms, and support the development of public health driven R&D initiatives, especially in limited-resource settings [17]. In 2022, MSF developed a clinical trial transparency policy, which guides the organization's efforts in making clinical trial costs public (Background in S1 Text) [18].

This study had two objectives: firstly, to analyze the costs of undertaking a pivotal clinical trial conducted in relatively low-resource health settings and, secondly, to demonstrate the feasibility of reporting clinical trial costs.

## Methods

We undertook a granular analysis of the monetary costs of the TB-PRACTECAL clinical trial from the trial sponsor perspective.

### Study setting

TB-PRACTECAL was a phase 2b-3 randomized controlled trial, TB-PRACTECAL, testing a new short, effective, and safe treatment regimen for drug-resistant TB. The trial was

conducted in Belarus, South Africa, and Uzbekistan (see Table A in S1 Text for a list of trial sites). The trial enrolled 552 patients across 6 sites. Enrolled patients were aged 15 years or older, with rifampicin-resistant pulmonary TB. Pregnant people were excluded from enrolment. (For full information on the recruited population, please see the main study publication [5]). The study protocol included 22 planned clinic visits over 108 weeks, alongside 3 sub-studies [5]. All trial visits took place in a hospital setting. The study included external quality monitoring for laboratory diagnostics and pharmacovigilance, and was performed in line with Good Clinical Practice. The trial was delivered with 19 partner organizations. Including planning, the project lasted 10 years.

## Expenditure data

Data on expenditures relating to the TB-PRACTECAL trial were collected from MSF accounting records, covering the period 1 January 2013 to 30 June 2023, 6 trial sites, and several implementing partners (Table A in S1 Text). Data were mostly available in a standard MSF bookkeeping template. For some sites, periods, and partners, different templates were used, or data were reported as an expense ledger, i.e., listing all individual transfers. Where needed, this was supplemented with review of contracts and invoices. As MSF was the funder and manager of the trial, as well as undertaking procurement, we believe that the collected data comprehensively cover direct trial expenditures (although certain indirect costs may not have been captured, as outlined in Limitations).

Extracted expenditure data were combined into one dataset in Microsoft Excel, resulting in a dataset of 3,483 individual expense items.

Figures were not inflation-adjusted, due to the relatively short period covered and difficulty in appropriately inflation-adjusting across at least 4 different countries. No discounting, risk-adjustment, or cost-of-capital adjustments were applied.

## Cost categorization and analysis

A list of cost categories was designed (Table B in S1 Text), with the simultaneous objectives of: sufficient granularity to enable useful insights; sufficient summarization to allow high-level overviews and cross-site comparisons; and sufficiently generalized categories to allow the list to be used in the future by MSF and other organizations as standardized categories for clinical trial cost disclosure and analysis. This process resulted in 27 cost categories (Table B in S1 Text). This was supported by a literature review of studies reporting clinical trial costs (Methods and Results in S1 Text, Tables C and D in S1 Text, Figure A in S1 Text).

Each expense item was allocated to one of the 27 defined cost categories by two reviewers; disagreements on categorization were resolved by consensus. To illustrate the data structure of the available bookkeeping data, expense items included, for example, a monthly salary payment for a specific individual; a procurement of a certain quantity of a medicine; payment for a mobile phone used in the field; payment for printing materials; the payment of an air fare.

During the categorization exercise, it was decided that it was not necessary to divide staff costs (or other costs) across several categories, for example, by making assumptions about the percentage of a staff member's effort dedicated to different functions. This was the case because a) the category definitions were such that the costs of individual staff members could be entirely attributed to one category; b) several functions were performed by a single external contractor or partner organization (e.g. data management, pharmacovigilance), who were paid in lump sum amounts. (While cost allocation methods such as step-down or 'ingredients' approaches were not required in this study, we recognize that they would likely be needed for analysis of other trials [19]).

Costs were analyzed by year, trial site, and category. In order to observe trends, costs were summated for each cost category (at two levels), year, and site. Costs for individual medical products were summated to identify the products with the highest cost. All expenditures associated with the trial found in the data sources were included in the analysis.

Expenditures were divided into central activities and site-level activities. (In some contexts, the former category may be referred to as 'sponsor costs' and the latter 'investigator costs'.) Central activities are the expenditures on the UK-based team that led the overall trial and provided various support to sites, as well as expenditures on services that supported all sites, including trial planning, day-to-day trial management, administration, statistical analysis, adverse effect monitoring, laboratory quality assurance and control, as well as electrocardiogram monitoring (required by the trial protocol due to anticipated cardiotoxicity risk). Site-led activities include principal investigator and other site staff costs, local regulatory compliance costs, trial site management, trial facility overheads, procurement of required equipment and consumables, diagnostics, and patient recruitment and follow-up.

## Per-patient costs

Average costs per-patient are a standard metric used for comparison of clinical trial costs across different contexts. We calculated both per-patient site-related costs and per-patient global costs. Per-patient site-related costs were calculated as the total expenses attributed to that clinical trial site divided by the number of patients enrolled at that site. Per-patient global costs were calculated as the sum of the per-patient site-related cost and the per-patient central cost (total central costs divided by the total number of patients enrolled in the trial).

In order to identify differences across sites, a 'heat map' was generated by comparing the cost of each activity at each site to the average across sites (S7 Table in S1 Text).

## Systematic review

In order to place our findings in context, a systematic review was undertaken to identify previous studies of clinical trial costs (Methods in S1 Text, Table C in S1 Text).

## Results

The overall cost of the TB-PRACTECAL program, including three sub-studies, was €33.9 million.

Central activities accounted for 26% total costs, site activities accounted for 72% of costs, and 2% of costs were uncategorizable (Table 1). 'Uncategorized costs' refers to those costs where the available bookkeeping data were not sufficient to assign them to a cost category. Among central activities, trial monitoring was the largest sub-category (10% of total trial costs) followed by trial management (10% of total trial costs)(Table 2). Among site activities,

**Table 1. High-level cost breakdown (EUR); a more detailed breakdown is shown in Table 2.**

| High-level cost category | Expenditure |
|---|---|
| Trial site staff costs | €10,849,002 (32%) |
| Central activities | €8,825,285 (26%) |
| External services supporting work at clinical trial sites | €5,203,370 (15%) |
| Purchase of materials | €5,212,607 (15%) |
| Other | €3,152,341 (9%) |
| Uncategorizable | €683,250 (2%) |
| **Total** | **€33,925,855** |

**Table 2. Breakdown of TB-PRACTECAL trial costs (EUR).**

| Cost category | Centrally procured | Belarus | Uzbekistan | | South Africaª | | Total (percentage of column) |
|---|---|---|---|---|---|---|---|
| | | Minsk | Karakalpak-stan | Tashkent | THINK | Wits | |
| **Trial site staff costs** | | | | | | | **€10,849,002 (32%)** |
| Trial site staff (specifically contracted) | €0 | €913,373 | €4,938,578 | €1,337,462 | €3,368,200 | €291,389 | €10,849,002 (32%) |
| **External services supporting work at clinical trial sites** | | | | | | | **€5,203,370 (15%)** |
| External clinical procedures | €17,244 | €259,564 | €0 | €5,540 | €135,233 | €0 | €417,581 (1%) |
| External diagnostics | €1,004,174 | €526 | €180 | €4,594 | €1,298,543 | €329,474 | €2,637,491 (8%) |
| External non-medical services | €1,390,710 | €11,145 | €0 | €28,360 | €12,204 | €0 | €1,442,418 (4%) |
| Funding of partner organization, not divisible into functions | €406,692 | €0 | €299,189 | €0 | €0 | €0 | €705,881 (2%) |
| **Purchase of materials** | | | | | | | **€5,212,607 (15%)** |
| Medical consumables (excl. medicines and vaccines) | €0 | €62,811 | €1,527,279 | €188,028 | €18,569 | €6,397 | €1,803,083 (5%) |
| Medical durables | €0 | €60,354 | €203,865 | €75,616 | €20,195 | €0 | €360,029 (1%) |
| Medicines and vaccines | €0 | €467,808 | €758,633 | €226,655 | €694,558 | €178,966 | €2,326,620 (7%) |
| Non-medical consumables | €50,127 | €37,548 | €259,164 | €12,876 | €68,954 | €0 | €428,668 (1%) |
| Non-medical durables | €17,560 | €14,528 | €16,473 | €52,065 | €193,581 | €0 | €294,207 (1%) |
| **Other** | | | | | | | **€3,152,341 (9%)** |
| Banking and tax | €0 | €3 | €0 | €0 | €7,309 | €0 | €7,311 (0.02%) |
| Community engagement | €136,065 | €4,861 | €0 | €3,858 | €1,144 | €5,407 | €151,335 (0.4%) |
| Facility operating costs | €38,073 | €18,316 | €125,485 | €13,789 | €184,613 | €28,398 | €408,674 (1%) |
| Losses, theft, expiries | €101,803 | €4,324 | €7,446 | €1,393 | €0 | €0 | €114,966 (0.3%) |
| Miscellaneous | €235,714 | €0 | €27,309 | €0 | €612,374 | €33,893 | €909,290 (3%) |
| Transport and travel | €432,200 | €122,265 | €699,644 | €220,865 | €80,753 | €5,038 | €1,560,766 (5%) |
| **Central activities** | | | | | | | **€8,825,285 (26%)** |
| Trial monitoring | €3,550,957 | – | – | – | – | – | €3,550,957 (10%) |
| Trial management | €3,304,868 | – | – | – | – | – | €3,304,868 (10%) |
| Trial planning | €1,253,698 | – | – | – | – | – | €1,253,698 (4%) |
| Data management | €317,475 | – | – | – | – | – | €317,475 (1%) |
| Regulatory compliance | €138,615 | – | – | – | – | – | €138,615 (0.4%) |
| Pharmacovigilance (safety reporting) | €63,665 | – | – | – | – | – | €63,665 (0.2%) |
| Central activities - not allocable to above categories | €7,841 | – | – | – | – | – | €7,841 (0.02%) |
| Analysis of results, publication | €188,167 | – | – | – | – | – | €188,167 (1%) |
| **Uncategorizable** | €683,250 | €0 | €0 | €0 | €0 | €0 | **€683,250 (2%)** |
| Total trial site costs | | | | | | | €33,925,855 |

ªTHINK (TB & HIV Investigative Network), in partnership with MSF, managed trial sites in Durban and Pietermaritzburg as a partner to MSF. Wits (Wits Health Consortium), in partnership with MSF, managed trial sites in Durban.

costs were driven by staff costs (32% of total trial costs), externally sourced diagnostic services (8%), medicines (7%), and other medical consumables (5%)(Table 2, Fig 1). Costs peaked in 2019 (year 3 of 6 active trial years) (Fig 2). Fifteen percent of costs were incurred before the start of enrollment, while 6% of costs were incurred after the last follow-up patient visit.

For costs incurred by the six clinical trial sites, the mean per-patient cost was €37,326 (range €19,998–45,942). When the cost of centrally-procured items and services, and central

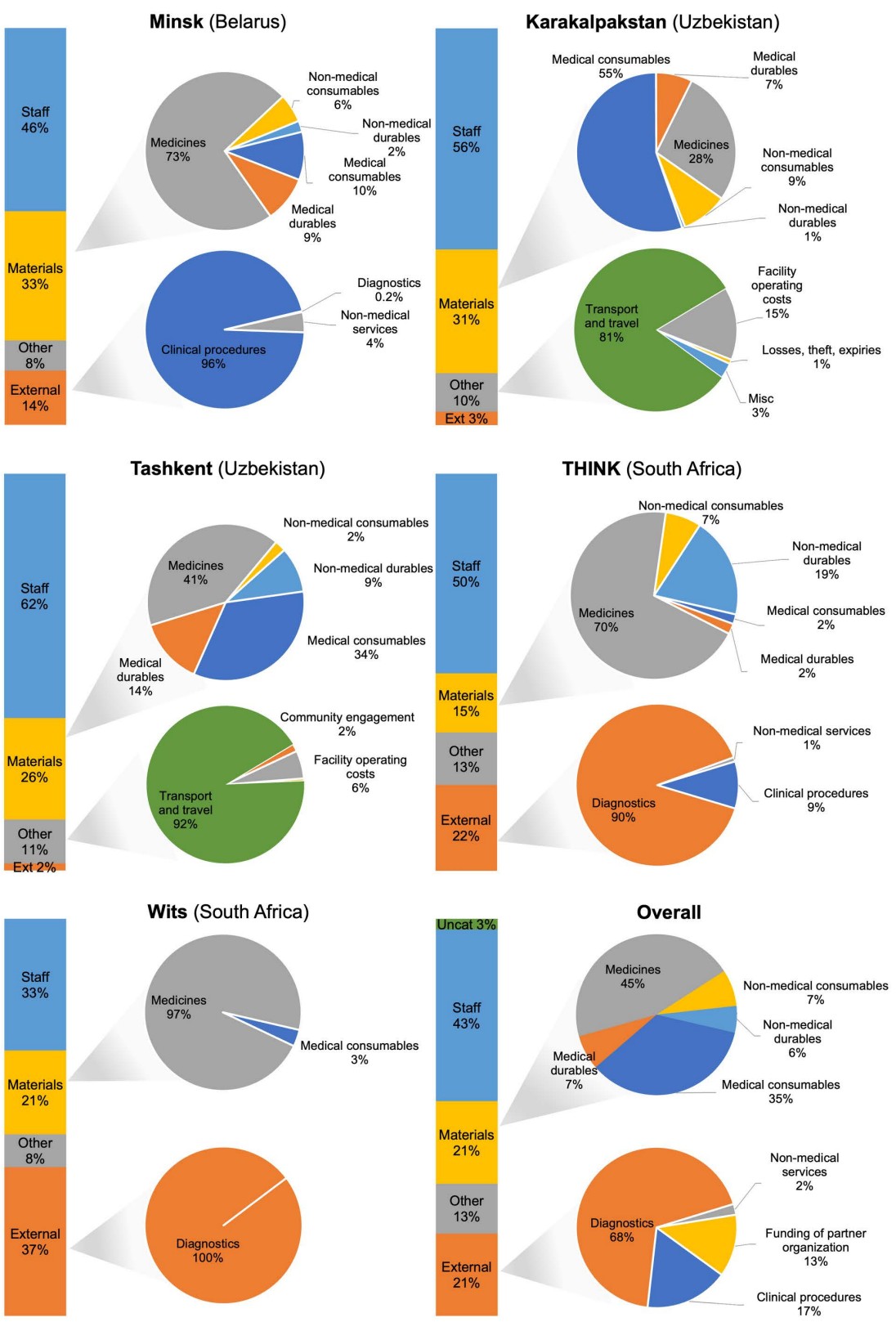

**Fig 1. Trial costs by site and cost category.**

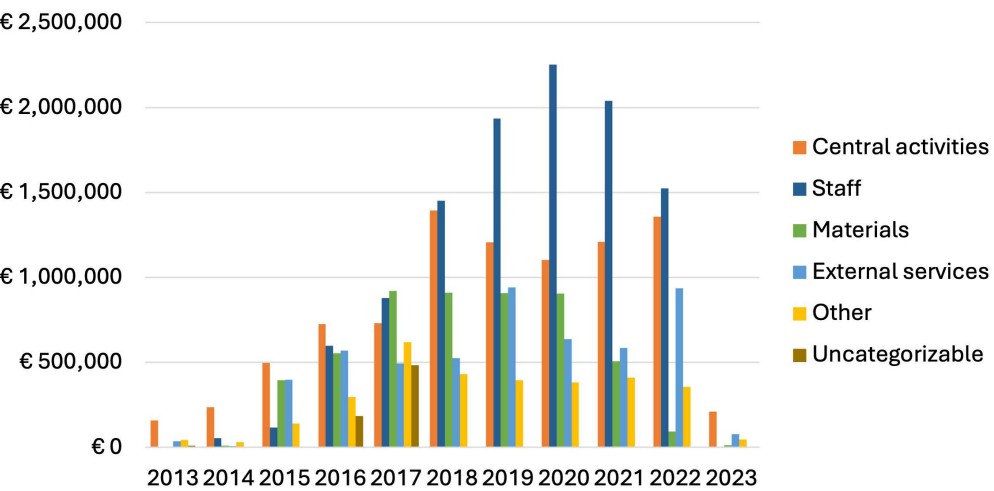

**Fig 2. Summary of trial costs by year and cost category.**

activities (e.g. trial management), are added, the mean global per-patient cost was €61,460 (range €44,132–70,076)(S5 Table). In the TB-PRACTECAL trial, 552 patients were enrolled and 410 patients completed the entire trial protocol; the per-patient averages presented here are based on the number enrolled at the start of the trial [5].

Top medicines by cost were bedaquiline (46% of all medicine costs), linezolid (16%), imipenem/cilastatin (10%), delamanid (9%), and clofazimine (4%)(S8 Table). Medicines in the investigational regimens (bedaquiline, pretomanid, linezolid, moxifloxacin, clofazimine) accounted for 68% of overall medicines costs (medicines costs broken down by individual medicines were available for Uzbekistan and Belarus, representing 63% of enrolled patients). The mean cost of medicines per patient was €4,215 (S6 Table). Most pretomanid used in the trial was donated by TB Alliance [5]; however, logistical issues in one site required additional pretomanid to be bought at cost. No other medicines were donated for use in the trial.

The systematic review identified 4,554 records, of which 22 full text articles met the inclusion criteria. Across the 22 publications (Table D in S1 Text), none reported a detailed empirical analysis of all costs incurred in a specific clinical trial. Of the 5 publications that reported average or estimated costs for phase 2 and 3 RCTs for pharmaceuticals, reported costs per trial ranged US$7–221 million (Results in S1 Text).

## Discussion

This study is, to our knowledge, the first publicly available granular reporting of the costs of a randomized clinical trial (RCT). We found that overall costs for TB-PRACTECAL, a pivotal trial for TB, were €34 million.

Previous analyses of pharmaceutical clinical trial costs report averages across groups of trials or provide modelled estimates, lacking the cost breakdowns required to understand cost drivers [11–13]. In our systematic review, studies published in the last 20 years report or estimate the costs of phase 2 and phase 3 pharmaceuticals RCTs in the range $7–221 million per trial (2023 USD; Methods in S1 Text). An analysis using 2004–2012 data from a commercial database of US clinical trial costs reported an average costs of US$23 million for anti-infective and respiratory phase 3 clinical trials [11]. A 2018 analysis modelled pivotal RCT costs using a commercial database of trial component costs, estimating a mean cost of $48.9 million for

clinical trials with an active-drug control arm, $51.7 million for clinical trials with a treatment duration longer than 26 weeks, and $64.7 million for a trial utilizing a clinical endpoint (rather than a surrogate outcome) [12]. The Drugs for Neglected Diseases initiative (DNDi), a non-profit organization that coordinates pharmaceutical R&D for neglected diseases, has published its clinical development costs, aggregated as total pre-clinical, phase 1, and late-phase costs (but not reporting costs of individual trials) [20].

The overall cost of TB-PRACTECAL, at €34 million, is within the region of the average costs for pivotal clinical trials reported in these studies. However, comparability is limited due to numerous factors, including clinical trial type (e.g., pharmaceutical development trials versus clinical management trials), enrolment number, and duration, among others [12,13,21–25]. On the one hand, one might expect lower trial costs when run in middle-income countries (such as Belarus, South Africa, and Uzbekistan) compared to high-income countries such as the US, due to lower labor and facility costs [26]. On the other hand, for some sites, TB-PRACTECAL required substantial investments by MSF to create or strengthen clinical trial infrastructure. Examples include purchasing laboratory equipment, back-up energy generators, temperature control and air filtration systems, emergency life-support equipment, and TB ward refurbishment – infrastructure that would have been available as standard in a well-resourced clinical trial unit. A previous estimate has put the cost of establishing clinical trial infrastructure for a TB trial at US$1–2 million per site per year [27]. Additionally, the costs of medicines used in the trial amounted to €2.3 million (just over €4,000 per patient), while, in commercial trials, investigational medical products are typically provided free by the manufacturer. TB-PRACTECAL also included three sub-studies, focused on quality-of-life endpoints, cost-effectiveness, and pharmacokinetics/pharmacodynamics [6–8]. While we did not attempt to estimate the costs attributable to each sub-study, the costs of pharmacokinetics/pharmacodynamics studies have been reported at $0.6–$1.8 million [28].

Other factors that likely increased the costs of TB-PRACTECAL include the trial's duration and adaptive design. The duration of TB-PRACTECAL was around 5.5 years from the first enrolled patient to the last follow-up clinic visit, which is longer than reported median durations of Phase 2 or Phase 3 trials (2.9 years and 3.8 years, respectively) [29]. TB-PRACTECAL followed an adaptive Phase 2b-3 trial design, starting as a Phase 2b trial with three intervention arms, with the most successful intervention arm extended into a Phase 3 trial [5,7]. The adaptive design likely increased costs [30], as the combined Phase 2b-3 trial can be considered to be greater in scope than a single-phase clinical trial.

The costs of clinical trials differ between countries with different resource levels [26]. Certain elements, as suggested above, may cost more in lower-resource settings, such as if new trial infrastructure needs to be built. However, increasing clinical research capacity in low- and middle-income countries is an important step towards equity in health research. Where and when stable clinical research infrastructure is established, economies of scale can be expected to bring costs down in the long run, as more clinical trials are undertaken.

There was a 2.3-fold difference between the lowest per-patient site costs (€19,998, Minsk) and the highest per-patient site costs (€45,942, Karakalpakstan)(S5 Table). A number of factors may explain these differences. Both sites in Uzbekistan (Karakalpakstan and Tashkent) had the highest costs for staff, medical consumables, medical durables, and transportation compared to other sites (S6 Table, S7 Table). MSF purchased lab equipment for the Uzbekistan sites and, in Tashkent, covered the costs of renovating the clinical trial unit. Additionally, all staff in the Uzbekistan sites were TB-PRACTECAL trial staff, while staff at the other sites were a mix of dedicated TB-PRACTECAL staff and staff working on multiple trials or supporting regular clinical care.

Overall, we conclude that MSF was able to keep costs at an average-to-low level despite the investments needed to create or strengthen clinical trial infrastructure at the sites.

The costs of medicines represented a significant expenditure in TB-PRACTECAL, with an overall cost of €2.3 million or 7% of total trial costs, with bedaquiline making up 46% of total medicine costs (S8 Table). High prices for new TB medicines, mainly for bedaquiline, delamanid, and linezolid, have been an enduring barrier to their introduction in high-burden countries' health systems [31,32]. The main patent on bedaquiline, the highest-priced medicine in the BPaL(M) regimen, expired in 2023 [33]. Following advocacy efforts by MSF and TB survivors, the proprietor committed to not enforcing secondary patents in 134 low- and middle-income countries [34]. This has enabled generic bedaquiline to enter the market, and significant price reductions are expected [35,36]. While the costs of the BPaL(M) regimens are now around US$400 per treatment course for countries eligible for procurement through the Global Drug Facility [35], further reductions will be a key component in enabling increased access.

Private sector investments in TB drug development have been very limited, making up 10% of global TB R&D over 2018-22 [37]. Public sector and non-profit research organizations have played a leading role in the development of new TB treatment regimens and sponsored pivotal clinical trials [38–40], filling a vacuum left by a lack of commercial interest in TB medicines.

Previous studies of RCT costs report modelled estimates or averages across aggregated data and often cannot be scrutinized due to the underlying granular data being confidential [11–15,23,26]. As a result, the quality of the data is impossible to assess, relative cost drivers cannot be identified, and costs cannot easily be compared between different trials. An improved understanding of clinical trial costs could inform medicines pricing policies, the design of R&D programs, and innovative R&D financing mechanisms [17]. Transparency in R&D costs is particularly pertinent for products whose development received public funding, to ensure accountability for the cost-effective use of those public funds. Wider, standardized reporting of clinical trials costs would support policymakers in crafting policy that balances access, through fair pricing, and innovation concerns. Increased transparency in clinical trial costs would also help in trial budgeting, grant applications and allocation, especially for non-profit or publicly financed drugs and clinical trials outside of high-income countries: the lack of data posed a challenge when TB-PRACTECAL was being planned [41].

Recent public legislative initiatives have also supported the move toward greater R&D cost transparency. For example, laws passed in some US states require the disclosure of R&D costs to justify price increases above a certain threshold, and several legislative proposals have been made to implement clinical trial cost transparency at the federal level [42], building on research into how clinical trial cost transparency could be implemented by the US NIH [10]. A 2021 European Parliament resolution called for the development of wide-ranging requirements for greater transparency in R&D costs as part of the EU Pharmaceutical Strategy [43]. A bill has been introduced to the Federal Senate of Brazil that would require drug manufacturers to disclose the costs of R&D, including the costs of clinical trials, as a requirement for product registration [44].

The pharmaceutical industry has long argued that high prices for medical products are required to recoup high R&D costs and sustain future innovation. However, research has shown that high drug prices are not justified by industry's spending on R&D [45,46]. Despite this, industry R&D estimates are often used to inform R&D policy and drug pricing debates. For clinical trials specifically, the lack of granular, publicly available data on costs has contributed to asymmetrical information in such debates. For example, the clinical trial cost data used in the US Congressional Budget Office's model estimating the impact of the Inflation Reduction Act (IRA) on future innovation was average clinical trial costs sourced from a

confidential, industry-sponsored survey. Access to granular clinical trial cost data would enable microsimulation and result in more precise and accurate models estimates [47,48]. Commercial secrecy around clinical trial costs may also have spillover effects on even not-for-profit actors: high and uncertain estimates may deter public or non-profit actors considering the financial feasibility of conducting clinical trials, especially in low-resource settings.

Tools to facilitate clinical trial cost reporting have been proposed previously, albeit specific to certain contexts [10]. In order to categorize TB-PRACTECAL costs, we developed a list of cost categories (Table B in S1 Text), guided by the objective of making cost categories sufficiently granular to allow detailed analysis while being sufficiently broad to provide an intuitive overview and allow comparison across different studies. MSF is building on the experiences of this analysis to design a general tool for reporting clinical trial costs, including for trials undertaken in humanitarian contexts. Reporting standards could be developed to facilitate analysis and comparability of economic data, similar to how the CONSORT (CONsolidated Standards Of Reporting Trials) checklist has improved comparability of clinical data [49].

## Limitations

Some costs were not captured, such as staff time in MSF central procurement operations and organizational overheads for the MSF UK team managing the trial. We expect that these costs, if captured, would be small compared to total trial expenditures.

The system of cost categorization used in this analysis does not separate costs by clinical function; for example, we have not attempted to identify the costs related to management of adverse effects. Cost categories were applied to accounting data retrospectively, which required minor compromises to maximise allocation of account lines to categories. Data on medicines costs were available for only two of the three countries included in the trial.

Generalizability of our findings is limited by the particular context in which the TB-PRACTECAL trial was performed. Further, the adaptive Phase 2b-3 clinical trial design of TB-PRACTECAL cannot be directly compared to typical Phase 2 or Phase 3 trials.

The TB-PRACTECAL trial had an incomplete follow-up rate of 6% [5]. The per-patient average costs are calculated based on the number enrolled, following a common approach [12,50]. However, this could be considered to 'deflate' the true per-patient average by up to 6%.

MSF acted as the funder and implementer of TB-PRACTECAL, and is the funder and implementer of this cost analysis study. Given this, there was, arguably, a risk of non-publication bias. This is the first MSF study for which publication of costs was planned. There may also be a perceived risk of bias in the direction of showing that MSF uses resources efficiently. However, as MSF is an independent organization funded by donations, receives research grants for trials infrequently, and the analysis was undertaken on finalized accounting records. The final total expenditure calculated by external consultants supporting the project (authors DG, MM, and MJB) was validated to match the total expenditure independently calculated for earlier interim accounting reports.

## Conclusion

This analysis is, to our knowledge, the first publicly available, granular reporting of costs for a clinical trial. The €34 million cost of TB-PRACTECAL included investments in developing clinical trial infrastructure, the complexity of managing six sites across three health systems, and medical expenditures that are not typical of standard clinical trials. Improved transparency in drug development costs, including clinical trials, is necessary to better inform

evidence-based pharmaceutical pricing policy attentive to access and innovation, as well as inform alternative drug development models, including non-profit and publicly led financing.

## Supporting information

**S1 Text. Appendix providing additional details on background, methods, and results.** (DOCX)

## Acknowledgements

We thank the participants of the TB-PRACTECAL trial.

We thank the experts who joined workshops to inform the methods and analysis: Ava Alkon (MSF), Candice Sehoma (MSF), Francisco Vegas (MSF), Nigel Masbayi (DNDi), Hassaan Zahid (MSF), Laura McCullagh (MSF), Laurence Vielfaure (DNDi), Christopher Morten (Columbia Law School), Rachel Cohen (DNDi), Olivier Wouters (LSE), Caleb Alexander (Johns Hopkins Bloomberg School of Public Health), Christophe Perrin (MSF), Jeanne Roussel (MSF) and Carole Mitnick (Harvard Medical School).

We thank Nicola James (MSF UK) for assistance in collecting MSF expenditure data. We thank Marion Conijn (MSF) for assistance in collecting data on medicines expenditures.

## Author contributions

**Conceptualization:** Dzintars Gotham, Manuel Martin, Melissa J Barber, Emil Kazounis, Rosalind Scourse, Greg Elder, Bern-Thomas Nyang'wa.

**Data curation:** Dzintars Gotham, Manuel Martin, Melissa J Barber, Emil Kazounis, Charlotte Batts.

**Formal analysis:** Dzintars Gotham, Manuel Martin, Melissa J Barber.

**Funding acquisition:** Rosalind Scourse, Greg Elder.

**Investigation:** Dzintars Gotham, Manuel Martin, Melissa J Barber.

**Methodology:** Dzintars Gotham, Manuel Martin, Melissa J Barber.

**Project administration:** Dzintars Gotham, Rosalind Scourse.

**Supervision:** Rosalind Scourse, Greg Elder, Bern-Thomas Nyang'wa.

**Writing – original draft:** Dzintars Gotham, Manuel Martin, Melissa J Barber, Rosalind Scourse.

**Writing – review & editing:** Dzintars Gotham, Manuel Martin, Melissa J Barber, Emil Kazounis, Rosalind Scourse, Greg Elder, Bern-Thomas Nyang'wa.

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
