## [Decision Letter · Decision Letter 0]

23 Jul 2024

PGPH-D-24-01202

Cost analysis of the TB-PRACTECAL clinical trial on novel tuberculosis treatment regimens

Dear Dr. Gotham,

Thank you for submitting your manuscript to PLOS Global Public Health. After careful consideration, we feel that it has merit but does not fully meet PLOS Global Public Health’s publication criteria as it currently stands. Therefore, we invite you to submit a revised version of the manuscript that addresses the points raised during the review process.

We look forward to receiving your revised manuscript.

Kind regards,

Raquel Muñiz-Salazar, Ph.D.

Academic Editor

Journal Requirements:

Additional Editor Comments (if provided):

The manuscript is well written and could transform the way trials are reported. Transparency about trial costs could lower drug prices and foster public-private partnerships in funding these studies.

However, the introduction lacks information and literature references, and the methodology is weakly presented. It is recommended to thoroughly reinforce the introduction and methodology before submission.

Reviewers' comments:

Reviewer's Responses to Questions

**Comments to the Author**

1. Does this manuscript meet PLOS Global Public Health’s publication criteria ? Is the manuscript technically sound, and do the data support the conclusions? The manuscript must describe methodologically and ethically rigorous research with conclusions that are appropriately drawn based on the data presented.

Reviewer #1: Yes

Reviewer #2: Partly

2. Has the statistical analysis been performed appropriately and rigorously?

Reviewer #1: N/A

Reviewer #2: Yes

3. Have the authors made all data underlying the findings in their manuscript fully available (please refer to the Data Availability Statement at the start of the manuscript PDF file)?

Reviewer #1: Yes

Reviewer #2: Yes

4. Is the manuscript presented in an intelligible fashion and written in standard English?

Reviewer #1: Yes

Reviewer #2: Yes

5. Review Comments to the Author

Reviewer #1: The study by Gotham et al is an analysis costs of the TB-PRACTECAL clinical trial on novel tuberculosis treatment regimens. This is an important study that sheds light on the costs and the major drivers of costs that are incurred during phase 2b-3 clinical trial to inform pricing of drugs. Overall, the manuscript is well written and could be a game changer in the way trials are reported. Openness about costs of trials may lower costs of drugs and foster Public Private participation in funding trials. The authors show that it is possible to keep trial cost at low levels despite long duration of trial, high cost of drugs used in the trial e.g. bedaquiline and extensive laboratory/facility investments that may be required in some settings. Congratulations to the authors.

Comments to the authors

Minor comments

Setting of the trial obviously influences the costs. In developed countries, the major cost drivers are HR costs, but in developing countries costs are pushed up due to setting up specialised facilities. Based on these results, what do authors suggest is the best way possible to lower the cost drivers? Perhaps the authors may make it clear that setting up trials in middle income countries even out health disparities and economies of scale can be realised in the long run if more trials are done. Openness of costs incurred during trials may foster collaborations by other players e.g. governments, NGOs, academia and pharmaceutical industry to lower costs further.

Attrition from the trial can have an impact on the costs. People who drop from the trial at the beginning (lower costs) than those who drop out around the end of the trial. Based on this, what was the % of participants who dropped out from the trial? S1 TABLE may be edited to reflect the number of participants that were enrolled/finished the trial.

Discussion:

Line 334: Based on the analysis of TB-PRACTECAL costs, we developed a standardized list of reportable cost categories (S2 Table). Is there scope to include costs that are associated with treatment/management of adverse events in S2 Table?

The text in Fig 1 is not very clear. May the authors upload files that have a high resolution.

May the authors use the referencing style where in text in in square brackets [].

Reviewer #2: Selected topic is interesting but lacking much information and literature in introductory chapter and very much weak presentation in methodology chapter extremely disappointed us. So, many relevant literature should be added to the introductory chapter with subtle references. Methodology is very vital chapter and issue of any empirical or secondary research. This manuscript has focused on discussion chapter rather than methodological areas/rooms ( Such as, study design/approach, study settings, population and sample, data processing/data collection and analysis and so on). Both "Introduction and methods" have been presented in very laconic way which undermined this manuscript.

6. PLOS authors have the option to publish the peer review history of their article (what does this mean? ). If published, this will include your full peer review and any attached files.

**Do you want your identity to be public for this peer review?** For information about this choice, including consent withdrawal, please see our Privacy Policy .

Reviewer #1: No

Reviewer #2: **Yes: ** Mohammad Ismail Bhuiyan

---

## [Decision Letter · Decision Letter 1]

20 Nov 2024

PGPH-D-24-01202R1

Cost analysis of the TB-PRACTECAL clinical trial on novel tuberculosis treatment regimens

Dear Dr. Gotham,

Thank you for submitting your manuscript to PLOS Global Public Health. After careful consideration, we feel that it has merit but does not fully meet PLOS Global Public Health’s publication criteria as it currently stands. Therefore, we invite you to submit a revised version of the manuscript that addresses the points raised during the review process.

We look forward to receiving your revised manuscript.

Kind regards,

Raquel Muñiz-Salazar, Ph.D.

Academic Editor

Additional Editor Comments (if provided):

The manuscript requires further revision before being considered for publication. The claim of using standardized cost analysis methods is unsupported, as established parameters for health services cost analysis already exist and can be readily adapted for clinical trial costing.

It is essential to address all the comments and suggestions provided by both reviewers. Incorporating their feedback will not only enhance the quality of the manuscript but also increase its chances of being accepted for publication.

Reviewers' comments:

Reviewer's Responses to Questions

**Comments to the Author**

1. If the authors have adequately addressed your comments raised in a previous round of review and you feel that this manuscript is now acceptable for publication, you may indicate that here to bypass the “Comments to the Author” section, enter your conflict of interest statement in the “Confidential to Editor” section, and submit your "Accept" recommendation.

Reviewer #3: (No Response)

Reviewer #4: (No Response)

2. Does this manuscript meet PLOS Global Public Health’s publication criteria ? Is the manuscript technically sound, and do the data support the conclusions? The manuscript must describe methodologically and ethically rigorous research with conclusions that are appropriately drawn based on the data presented.

Reviewer #3: Partly

Reviewer #4: Yes

3. Has the statistical analysis been performed appropriately and rigorously?

Reviewer #3: No

Reviewer #4: I don't know

4. Have the authors made all data underlying the findings in their manuscript fully available (please refer to the Data Availability Statement at the start of the manuscript PDF file)?

Reviewer #3: Yes

Reviewer #4: Yes

5. Is the manuscript presented in an intelligible fashion and written in standard English?

Reviewer #3: No

Reviewer #4: Yes

6. Review Comments to the Author

Reviewer #3: See attached file.

Reviewer #4: This study aims to analyze the costs of undertaking a pivotal clinical trial conducted in relatively low-resource health settings and to demonstrate the feasibility of reporting clinical trial costs, based on a multicenter two-stage adaptive phase 2b-3 randomized controlled trial, TB-PRACTECAL, testing a new treatment regimen for drug resistant TB in Belarus, South Africa, and Uzbekistan.

The article is well written and very well comprehensive, bringing up the call for transparency in the costs of R&D and in drug development to inform medicine pricing negotiations, a key element in the design and implementation of more equitable systems.

The following are my comments/suggestions:

- Would be interesting to know the loss to follow-up rate, and how this affected the financial efficiency of the trial.

- Line 95- R&D, write the abbreviation in full.

- Line 341- STROBE, write the abbreviation in full. Suggest including a description of this initiative.

- Check all the abbreviations across the manuscript.

- In the discussion section, the US and the European Parliament are mentioned as examples of public legislative initiatives supporting the move toward greater R&D cost transparency. Are there examples from low-income countries’ initiatives? Would be interesting to mention.

7. PLOS authors have the option to publish the peer review history of their article (what does this mean? ). If published, this will include your full peer review and any attached files.

**Do you want your identity to be public for this peer review?** For information about this choice, including consent withdrawal, please see our Privacy Policy .

Reviewer #3: No

Reviewer #4: No

---

## [Decision Letter · Decision Letter 2]

25 Feb 2025

Cost analysis of the TB-PRACTECAL clinical trial on novel tuberculosis treatment regimens

PGPH-D-24-01202R2

Dear Dr. Gotham,

We are pleased to inform you that your manuscript 'Cost analysis of the TB-PRACTECAL clinical trial on novel tuberculosis treatment regimens' has been provisionally accepted for publication in PLOS Global Public Health.

Best regards,

Raquel Muñiz-Salazar, Ph.D.

Academic Editor

Both reviewers have confirmed that the authors have carefully addressed all suggestions and comments provided during the review process.

As all requirements have been satisfactorily met, your paper has been accepted for publication.

Reviewer Comments (if any, and for reference):

Reviewer's Responses to Questions

**Comments to the Author**

1. If the authors have adequately addressed your comments raised in a previous round of review and you feel that this manuscript is now acceptable for publication, you may indicate that here to bypass the “Comments to the Author” section, enter your conflict of interest statement in the “Confidential to Editor” section, and submit your "Accept" recommendation.

Reviewer #3: All comments have been addressed

Reviewer #4: All comments have been addressed

2. Does this manuscript meet PLOS Global Public Health’s publication criteria ? Is the manuscript technically sound, and do the data support the conclusions? The manuscript must describe methodologically and ethically rigorous research with conclusions that are appropriately drawn based on the data presented.

Reviewer #3: Partly

Reviewer #4: Yes

3. Has the statistical analysis been performed appropriately and rigorously?

Reviewer #3: Yes

Reviewer #4: N/A

4. Have the authors made all data underlying the findings in their manuscript fully available (please refer to the Data Availability Statement at the start of the manuscript PDF file)?

Reviewer #3: Yes

Reviewer #4: Yes

5. Is the manuscript presented in an intelligible fashion and written in standard English?

Reviewer #3: Yes

Reviewer #4: Yes

6. Review Comments to the Author

Reviewer #3: The authors seem to have addressed the review comments provided earlier.

Reviewer #4: (No Response)

7. PLOS authors have the option to publish the peer review history of their article (what does this mean? ). If published, this will include your full peer review and any attached files.

**Do you want your identity to be public for this peer review?** For information about this choice, including consent withdrawal, please see our Privacy Policy .

Reviewer #3: No

Reviewer #4: No
